# An Evaluation of Staff Engagement with Infectious Healthcare Waste Management Policies: A Case Study of Tunisia

**DOI:** 10.3390/ijerph17051704

**Published:** 2020-03-05

**Authors:** Kaouther Maaroufi, Terry Tudor, Mentore Vaccari, Afef Siala, Ezzeddine Mahmoudi

**Affiliations:** 1Department of Life Sciences, Faculty of Sciences of Bizerte, University of Carthage, Jarzouna 7021, Tunisia; ezzeddine.mahmoudi@gmail.com; 2Department of Environmental and Geographical Sciences, The University of Northampton, Waterside Campus, Northampton NN1 5PH, UK; terryl.tudor@gmail.com; 3Department of Civil, Architectural and Environmental Engineering and of Mathematics, University of Brescia, Via Branze 43, 25123 Brescia, Italy; mentore.vaccari@unibs.it; 4National Agency for Waste Management, 19 Rue de Jerusalem Street, 2 Tunis 100, Tunisia; gidf@anged.nat.tn

**Keywords:** developing countries, healthcare waste practices, medical waste, infection prevention and control

## Abstract

This study evaluated the engagement of staff regarding infectious healthcare waste management, in two case-study universities in Tunisia. Using a questionnaire survey, it was found that the most significant reported factors that influenced engagement were the availability of technical sheets and posters, training and education programs, and the age range of the staff. While there was some accordance with the Tunisian Decree application *n*° 2008–2745 of July 28th, 2008, as well as international guidelines and best practice (e.g. the use of color coded bins, waste management teams, and infection control measures), there were also limitations in the provision of training. This limitation in training and, to a lesser extent, awareness impacted on the beliefs about infectious healthcare waste management of staff and their practices. Recommendations for addressing these issues are suggested.

## 1. Introduction

Healthcare activities generate infectious healthcare waste (IHCW), which poses risks for humans and the environment. Considering the potential risks, attention should be given to the management of IHCW. However, a recent joint WHO/UNICEF evaluation found that just over half (58%) of the facilities in low- and middle-income countries had inadequate systems for the safe disposal of the waste [1]. Although the amount of IHCW is minimal compared to the total amount of waste generated, there is growing concern for its management [2]. For example, the levels of knowledge and attitudes, as well as the waste management practices of clinical staff in developing countries, are often low [3,4].

Various studies have indicated that effective management of the waste should take account of certain key factors, including written policies and clear guidelines [5], the involvement of different stakeholders in system development planning [6], the managerial and organizational structure [7], operational training for staff at different levels [8], suitable equipment for waste handling and protection tools [9], appropriate treatment options [10,11] and operation [8], and the existence of qualified waste managers [12]. In addition, the management of IHCW should be in accordance with procedures in (international) guidance for infectious healthcare waste management (IHCWM) [13]. Procedures should take account of all the operational stages of the handling, disposal, and treatment processes. To reduce the risk of the transmission of infections, there should be effective infection control measures such as effective hand washing and management of ICHW in place [14]. However, studies undertaken in various low – and middle-income countries have found that, generally, levels of knowledge of infection control measures were limited and practices did not meet the standards [15,16,17,18,19,20].

Using Tunisia as the case-study country, this research aimed to determine the beliefs, levels of knowledge and practices regarding IHCWM and the key factors that influenced these issues amongst staff.

In Tunisia, the decree of application *n* ° 2008–2745 dated July 28th, 2008 mandated the creation of waste management units in each health care facility [21]. The decree strongly recommended the sharing of information and use of training to effectively manage IHCW, through, for example, more efficient waste segregation. The decree also mandated the use of appropriate color coded bins and storage facilities to facilitate effective waste segregation by nursing staff. Indeed, a national Tunisian standard was adopted in 2015—NT 106.85–106.93, regulating the packaging of IHCW [22]. However, to date there has been limited empirical evaluation of the effectiveness of the decree or of IHCWM generally in Tunisia.

## 2. Materials and Methods

### 2.1. Setting

This study was performed in two university hospitals (Table 1). These sites were chosen because they are classified in Tunisia as pilot establishments regarding service care quality and organizational structure. Both public facilities are located in Grand Tunis. Grand Tunis is in the greater metropolitan area in the north of Tunisia. Approval for the study was granted at the institutional level, following a request from the National Agency of Waste Management. 

A combination of factors was employed to identify the department to be surveyed, including those with the highest potential IHCW production rates [23], and potential contamination risks [24]. Based on these criteria, three departments were chosen, namely, surgical, emergency, and intensive care.

A questionnaire survey was conducted from March to September 2018. The questionnaire contained both closed-ended and open-ended questions. It was comprised of 13 questions, including sociodemographic characteristics, knowledge and practices regarding infectious waste management, attendance at training programmes, and engagement with infection control measures. It specifically aimed to assess the knowledge of staff about IHCW management and infection control, as well as to understand their waste management and infection control practices, and any key influencing factors (Appendix A).

### 2.2. Subject

The following criteria were employed to choose the staff from the three departments:

Inclusion criteria:

Clinical staff, namely, doctors, surgeons, senior technicians, caregivers, nurses, midwives, as well as agents and people who handle infectious waste.

Exclusion criteria:

Any person who met the inclusion criteria but was absent during the survey or refused to be investigated.

A total of 492 questionnaires were distributed at the two sites, comprised of 167 questionnaires (Site 1) and 325 questionnaires (Site 2). Arrangements were made with the supervisor of each department to receive the completed questionnaire from participants. Some 182 completed questionnaires were received from the two sites, representing a response rate of 39%, which is well within the range of similar surveys.

The analyses of the data were carried out from September 2018 to July 2019. It was undertaken in three main steps, namely,

−Descriptive analyses of the characteristic of the sites and staff;−Bivariate analyses, using Pearson correlations to determine significant influencing factors on policies and practices regarding IHCWM (the dependent variable: the site has measures to manage waste);−Hierarchical analyses to determine the strength of the link between the various factors on practices and beliefs associated with IHCWM.

## 3. Results

Table 2 shows that the sample was comprised of 20% doctors, 3% managers, 35% technicians, and 6% workers. The average age was 36 years old, and the average length of time working within the healthcare sector was 13 years.

### 3.1. Level of Awareness of Infectious Healthcare Waste Management Documents

Figure 1 illustrates that awareness of documents related to HCWM in general was low, at less than 30%, at both sites.

At site 1, staff were more aware of technical sheets and posters (34%) than guidance of good practices (28%). At site 2, staff were more aware of guidance of good practices (25.5%) than technical sheets and posters (18%). There were low levels of awareness of the manual of procedures: 15.9% for site 1, some 9.6% for site 2, and 12.6% on average for both sites.

On average, 75% of the staff believed that their department had a strategy to manage infectious waste. This percentage was 77% at site 1 and 73% at site 2.

### 3.2. Infectious Healthcare Waste Management Measures

Table 3 shows that the most reported measure for IHCWM was the use of equipment (51%) at both sites. The main equipment employed included the use of colored containers and bags for containment and disposal. At site 1, the most reported measures were use of technical sheets and posters (60%), and equipment (49%). For site 2, equipment was the most reported measure (53%).

Table 4 indicates that the three most cited strategies for managing the waste were (1) the availability of equipment (51%), (2) use of technical sheets and posters (48%), and (3) awareness and training on best practices related to IHCWM (41%).

### 3.3. Training

Figure 2 illustrates that for most staff (68%), waste management was perceived to be a major concern, and most were of the view that training on IHCWM was important (71% site 1 and 66% in site 2). However, only 44% of staff had attended such training (43% at site 1 and 38% at site 2).

At site 1 and site 2, the most undertaken measures for infection control were hand washing (81.8%) and (88%) and use of alcohol gel (76%) and (86%), respectively (Table 5).

Table 6 lists the three most cited strategies for infection control utilized, which were hand washing (85%), (2) use of alcohol gel (81%), and (3) the cleaning of rooms and equipment (56%).

### 3.4. Key Influencing Factors

Table 7 shows that there was a correlation between the belief of staff that their department had a strategy for waste management and age rage (*p* < 0.01), the existence of technical sheet and posters (*p* < 0.01), the awareness and training on good practices of HCWM (*p* < 0.01), and the use of alcohol gel as an infection control measure (*p* < 0.05).

There were four key factors that, combined, explained approximately 54% (32% + 21.8%) of the issues stated (Table 8). Within these factors, there were four main issues that influenced the beliefs and behavior of staff on IHCWM practices, namely, (1) the use of technical sheets and posters, (2) awareness and training on good practices of waste management, (3) use of alcohol gel: hygiene and cleaning practices, and (4) the age range of staff (i.e. the length of time working within the healthcare sector). Table 9 lists the five different age groups which were considered in the study. 

## 4. Discussion

### 4.1. Key Waste Management and Infection Control Practices Employed

The most reported measure for IHCWM was the use of equipment (51%) at both sites. On average, the main type of ‘equipment’ employed included the use of color-coded containers and bags. This was in accordance with international guidelines [13], and best practice in other countries e.g., [3,24,25]. In Tunisia, color-coded containers became mandatory following the decree [21]. Thus, elements of the decree do appear to have been implemented on the ground. Both sites were also contracted with a transport and waste treatment company as a means of ensuring best practice. In Tunisia, it is mandatory for transport personnel to wear appropriate personal protective equipment (PPE), including gloves, closed shoes, overalls, and masks [26].

At site 1, the most reported measures were the use of technical sheets and posters (60%) and equipment (49%), and at site 2, the most reported measure was the use of equipment (53%). While there were minimal differences between the two sites, the use of the technical sheets and posters at site 1 led to higher awareness and more effective practices. Both sites had a healthcare-waste team manager. For site 1, the waste team was comprised of one hygienist and 30 staff such as cleaners and waste workers. While at site 2, the team was comprised of two doctors, one veterinarian technician (specialized in the environment and hygiene), and four waste workers who specialized in the collection of healthcare waste from different departments of the facility. The presence of these teams was crucial to effective practice. Again, the existence of the teams, as mandated by the decree, indicates that when implemented, the decree facilitated more effective IHCWM.

With respect to infection control, the most undertaken measures were hand washing (81.8%—site 1, and 88%—site 2) and the use of alcohol gel (76%—site 1, and 86%—site 2). These measures are in keeping with international standards [13], and best practice in other countries, e.g., [3,27].

### 4.2. Key Factors that Influenced the Practices

There were two key factors that, combined, explained approximately 54% of the issues stated by the staff. Within these factors, there are four main issues that influenced the beliefs and behavior of staff on IHCWM practices, namely: (1) the use of technical sheets and posters, (2) awareness and training on good practices of waste management, (3) use of alcohol gel: hygiene and cleaning practices, and (4) the age range of staff (i.e., the length of time working within the healthcare sector). These results suggest that the more aware and trained staff were, and the longer they had worked at the site, the more competent they were to undertake effective management of the IHCW. Age was found to have been a significant factor that influenced the beliefs and practices of the staff at both sites. Similar to [28] but contrary to [29], staff who had more years of service also had stronger beliefs and more effective IHCWM practices. Overall, therefore, similar to previous studies undertaken in developing countries, e.g., [3,14,25,30,31,32], the provision of adequate training, as well as awareness building, are crucial for effective IHCWM.

On average at both sites, most staff (75%) were of the view that their site had policies and a strategy in place for the management of the waste. However, despite both sites having documents in place on best practice for IHCWM, there was limited awareness about the documents. A possible reason for this might have been that access to these documents was limited to the hygiene team. In addition, while there was concern about management of the waste and a willingness to attend training, attendance at such was generally limited. However, this perceived dichotomy in beliefs and practice can be explained in part by the fact that opportunities to access training were limited, particularly at site 2. The importance of training is similar to other findings, e.g., [3], but in contrast to others, e.g., [28]. The development of a training programme should take account of all subjects of IHCWM, including effective waste containment, segregation and management, infection prevention and control, and staff roles and responsibilities.

### 4.3. Barriers and Limitations

There were various logistical barriers at the two sites that in some cases limited the capacity and the implementation of procedures by the waste committee team. These barriers included the time and resources that could be dedicated by and for staff. Some 2% of staff also mentioned limitations in the existence of an authorized facility for IHCW treatment. In 2015, there was a strengthening of the regulatory framework by the promulgation of a Joint Order of the Minister of the Environment and Health of June 6, 2014, with set requirements contained in the agreement concluded between the healthcare establishment and the authorized facility of IHCW treatment [33]. Thus, the number of the healthcare establishments contracted with private companies for the collection, transport, and treatment of IHCW increased from 15 facilities in 2014 to 83 facilities in 2017.

There were some limitations in this study, including time and access. The time for the granting of the necessary permissions and some logistical issues limited access to staff and data collection. Future studies should therefore incorporate a wider number of sites and staff in order to have a more holistic perspective of the manner in which IHCW is being managed across Tunisia. 

## 5. Conclusions

This study is one of the first empirical studies to examine the perceptions, beliefs, attitudes, and practices of staff regarding IHCWM in Tunisia.

While there was some accordance with the Tunisian Decree, as well as international guidelines and best practice (e.g., the use of color coded bins, waste management teams, and infection control measures), there were also limitations in the provision of training. This limitation in training and, to a lesser extent, awareness impacted beliefs about IHCWM of staff and their practices. Thus, there is a need to focus efforts on the provision of training and awareness building for all clinical and nonclinical staff, at all levels (including senior managers), to improve effectiveness of IHCWM in Tunisia. Efforts to improve knowledge, beliefs and practices rely on the continuity of sensitization activities; training; and the availability of operational resources, which are the factors of success of IHCWM. The increase in the number of private contractors for the disposal and treatment of IHCW was related to the development of financial and operational resources dedicated to IHCWM. The creation of a separate budget line to cover the costs of the IHCWM since 2013 has been a key factor in the improvement of the IHCWM and its durability over time. This issue was a strategic driver for sustainable development within IHCWM in Tunisia. In addition, it is vitally important to strengthen the control and inspection mechanisms of waste management operations. Finally, at the overarching level, it is vitally important for the health ministry and National Agency for Environment Protection to rigorously supervise waste disposal.

## Figures and Tables

**Figure 1 ijerph-17-01704-f001:**
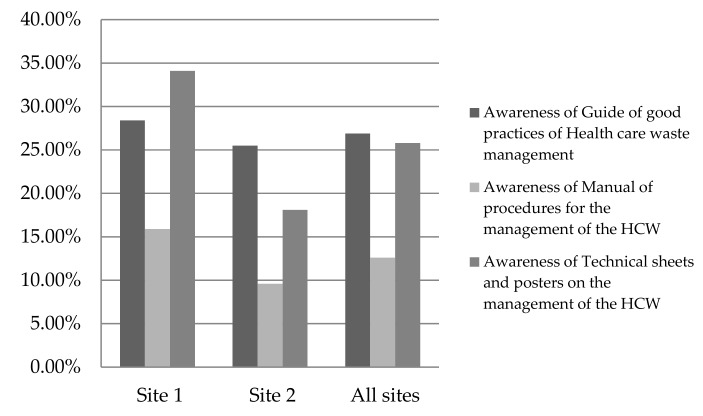
Staff knowledge about documentation related to infectious healthcare waste management.

**Figure 2 ijerph-17-01704-f002:**
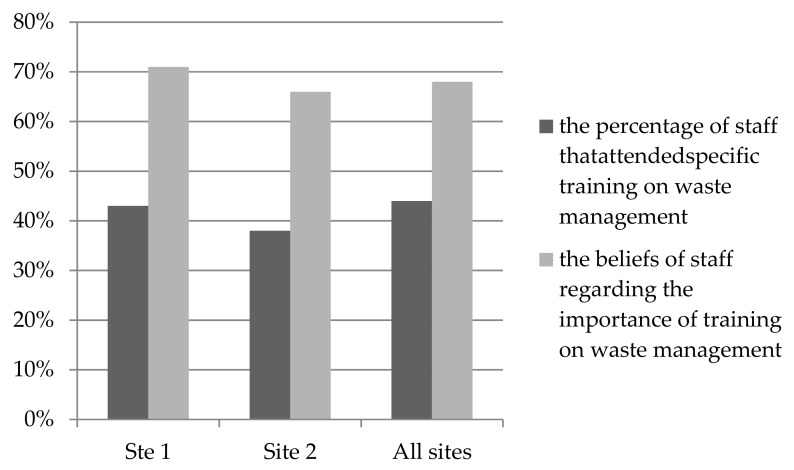
Beliefs regarding the importance of training and percentage of staff attendance at training.

**Table 1 ijerph-17-01704-t001:** The study area.

Sites	Site 1	Site 2
Governorate	Tunis	Tunis
Number of services	16	34
Bed number	344	600
Occupancy rate	80.6%	100%
The yield of healthcare waste	0.63 kg/bed/day	0.63 kg/bed/day
Waste management unit team	One Hygienist 30 waste workers	Two doctors 1 veterinarian technicians (specialized in the environment, hygienist) Four waste workers

**Table 2 ijerph-17-01704-t002:** Socio-demographic characteristics of the respondents.

Subject Characteristic	Job Role	Interviewees by Department
Female 58%	Doctors 20%	Emergency 23%
Male 40%	Manager 3%	Intensive care 20%
Average age: 36 years old	Nurse 33%	Surgery 48%
Average length of time working within the healthcare sector: 13 years	Technicians 35%	Not mentioned 9%
	Workers 6%	
	Not mentioned 3%	

**Table 3 ijerph-17-01704-t003:** Comparison of measures reported by staff regarding infectious waste management.

Measures	Frequency of Response Site 1	% Site 1	Frequency of Response Site 2	% site 2
Equipment	43	49 %	50	53%
Technical sheet and posters	53	60%	34	36%
Awareness and training	38	43%	36	38%
Existence of a committee on waste management (unit)	22	25%	30	32%
Treatment company	2	2%	2	2%

**Table 4 ijerph-17-01704-t004:** Strategies utilized for waste management.

Measures Reported to Manage Waste at Both Sites	Frequency of Response at Both Sites	% Both Sites
Equipment	93	51%
Technical sheet and posters	87	48%
Awareness and training	74	41%
Existence of a committee on waste management (unit)	52	29%
Treatment company	4	2%

**Table 5 ijerph-17-01704-t005:** Comparison of strategies reported by staff being undertaken at the sites for infection control.

Measures	Frequency of Response Site 1	% Site 1	Frequency of Response Site 2	% Site 2
Alcohol gel	67	76%	81	86%
Hand washing	72	81.8%	83	88%
Isolation chamber	29	33%	37	39%
Personal protective equipment	34	38.6%	45	47.9%
Cleaning rooms and equipment	48	54.5%	54	57%
Awareness and training	31	35%	38	40%
Audits	7	8%	15	16%

**Table 6 ijerph-17-01704-t006:** Strategies reported for both sites regarding infection control.

Measures	Frequency of Response Both Sites	% Both Sites
Alcohol gel	148	81%
Hand washing	155	85%
Isolation chamber	66	36%
Personal protective equipment	79	43%
Cleaning of rooms and equipment	180	56%
Awareness and training	69	37.9%
Audits	22	12%

**Table 7 ijerph-17-01704-t007:** Factors related to the staff belief that the department has a strategy of waste management.

Variables	Pearson Correlation	Sig.
Age range	−0.193	0.01
Technical sheets and posters	0.628	0.01
Awareness and training on good practices of HCWM	0.603	0.01
Alcohol gel	0.164	0.05

**Table 8 ijerph-17-01704-t008:** Factorial analysis regarding the beliefs of staff on key influencing factors.

Factor Matrix ^a.^	Factors	
	1	2
Technical sheet and posters	0.64	−0.19
Awareness and training	0.63	−0.28
Alcohol gel	0.62	0.49
Age range	−0.02	0.71
% Var	32%	21.8%

Extraction method: Factorization in main axes. ^a^ Attempt to extract two factors. More than 25 iterations are required. (Convergence = 003). The extraction was interrupted.

**Table 9 ijerph-17-01704-t009:** Age ranges of the participants.

Groups	Age Range
1	16–25
2	26–35
3	36–45
4	46–55
5	Over 56

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
