# Peer review of "An Evaluation of Staff Engagement with Infectious Healthcare Waste Management Policies: A Case Study of Tunisia"

_ijerph, 2020, doi:10.3390/ijerph17051704_

Round 1

Reviewer 1 Report

Journal:Int. J. Environ. Res. Public Health

Title: An evaluation of staff engagement with infectious healthcare waste management policies: a case study of Tunisia

The manuscript entitled " An evaluation of staff engagement with infectious healthcare waste management policies: a case study of Tunisia " study evaluated the engagement of staff regarding infectious healthcare waste management, in two case study universities in Tunisia. The manuscript appears to be well-organized, and the results and discussions are good. However, some corrections are also needed. I recommend major revision before the manuscript can be accepted for publication. I hope the specific comments below will help improve the quality of this manuscript.

Comments

  • Format. The format of whole manuscript should be improved. For example, Line 76, Table 1. Thestudy area→Table 1. The study area; Figure 2, the percentage of staffthatattendedspecific training on waste management→the percentage of staff that attended specific training on waste management.
  • The framework and English language of the paper should be improved.
  • The yield of healthcare waste and geographical situation should be set as the indexes to choose the hospitals.
  • Supply the yield of healthcare waste of hospitals in table 1.
  • Table 3 and table 4. How to calculate the percentage of Equipment and other?For example (table 3), the Frequency of Equipment in response Site 1 was 43, the percentage of Equipment was 49%. 49%=43/?
  • The data should be analysed in-depth.
  • Line 194. There are four main issues that influenced the beliefs and behavior of staff on IHCWM practices, including the age range of staff. Provide the number of different age groups.

Reviewer 2 Report

This paper needs a bit more commentary in the discussion section. The percentage of returned questionairres was low- any clear reasons why, and did return favor some groups/worker types etc. The paper would also greatly benefit from deeper discussion how to change this-for example is there an orientation for all new employees and at that time all new hires could be educated about the need to handle hospital waste properly. Last- this is a reach- is there any data or even general concern about things like needle sticks at the hospitals making the case for better worker knowledge and procedure following behavior. Also- authors might propose ideas how to improve following of the rules.

Reviewer 3 Report

This paper has some value in terms of questionnaire study that explains perceptions, beliefs, and attitudes of facility staff with infectious 2 healthcare waste management policies. 

Round 2

Reviewer 1 Report

It is to be noted that a large amount of work has been done to improve the manuscript. In general, I am satisfied with the responses given.